# Nanotechnology Frontiers in γ-Herpesviruses Treatments

**DOI:** 10.3390/ijms222111407

**Published:** 2021-10-22

**Authors:** Marisa Granato

**Affiliations:** Department of Experimental Medicine, Sapienza University of Rome, Piazzale Aldo Moro 5, 00185 Roma, RM, Italy; marisa.granato@gmail.com

**Keywords:** Epstein–Barr, EBV, Kaposi’s sarcoma associated-herpesvirus (KSHV), nanoparticles (NPs), virus like-particles, VLPs, autophagy, apoptosis, γ-herpesviruses

## Abstract

Epstein–Barr Virus (EBV) and Kaposi’s sarcoma associated-herpesvirus (KSHV) are γ-herpesviruses that belong to the *Herpesviridae* family. EBV infections are linked to the onset and progression of several diseases, such as Burkitt lymphoma (BL), nasopharyngeal carcinoma (NPC), and lymphoproliferative malignancies arising in post-transplanted patients (PTDLs). KSHV, an etiologic agent of Kaposi’s sarcoma (KS), displays primary effusion lymphoma (PEL) and multicentric Castleman disease (MCD). Many therapeutics, such as bortezomib, CHOP cocktail medications, and natural compounds (e.g., quercetin or curcumin), are administrated to patients affected by γ-herpesvirus infections. These drugs induce apoptosis and autophagy, inhibiting the proliferative and cell cycle progression in these malignancies. In the last decade, many studies conducted by scientists and clinicians have indicated that nanotechnology and nanomedicine could improve the outcome of several treatments in γ-herpesvirus-associated diseases. Some drugs are entrapped in nanoparticles (NPs) expressed on the surface area of polyethylene glycol (PEG). These NPs move to specific tissues and exert their properties, releasing therapeutics in the cell target. To treat EBV- and KSHV-associated diseases, many studies have been performed in vivo and in vitro using virus-like particles (VPLs) engineered to maximize antigen and epitope presentations during immune response. NPs are designed to improve therapeutic delivery, avoiding dissolving the drugs in toxic solvents. They reduce the dose-limiting toxicity and reach specific tissue areas. Several attempts are ongoing to synthesize and produce EBV vaccines using nanosystems.

## 1. γ-Herpesviruses

Epstein–Barr Virus (EBV) and Kaposi’s sarcoma associated herpesvirus (KSHV) are γ-herpesviruses belonging to the *Herpesviridae* family [1]. They are enveloped viruses, with double stranded DNA (130–150 Kbps) restrained in icosahedral capsids, surrounded by envelopes with several glycoproteins (GPs), which is necessary during the recognition and infection of the host cells. EBV Gp350 exploits the binding and the attachment to CR2–CD21 B cell receptors to infect these kinds of cells [2]. The γ-herpesviruses acquire the primary envelope at the nuclear membrane by the maturation steps, the primary envelopment-de-envelopment steps of the nuclear inner and outer leaves. They are conserved in all herpesviruses, such as herpes simplex type 1 (HSV1). The nuclear membrane complex, called NEC, is able to ‘gain’ the nucleocapsids—to the cytoplasmatic compartment—in which the herpesviruses acquire the last envelope and the glycoproteins require new viral infections.

In this review, I discuss the therapeutics that sustain γ-herpesvirus maturations in new viral particles. The mechanisms underlining the lytic cycle have also been explored by clinicians to improve the conventional therapy in patients affected by EBV- and KSHV-associated diseases.

### 1.1. Epstein–Barr Virus (EBV)

EBV infects 95% of the worldwide population; it is usually asymptomatic. However, several circumstances could induce microenvironments that are related to the onset of certain diseases. EBV is the etiological agent of infectious mononucleosis (IM), and it is associated with Burkitt lymphoma, nasopharyngeal carcinoma, a subtype of gastric cancer, and lymphoproliferative disorders, often arising in solid organ transplantation patients (PTDLs) [3,4,5,6,7,8]. In the last decade, EBV infection has also been related to the onset of immunological disorders, such as multiple sclerosis and rheumatoid arthritis [8]. Infection of B cells is a reservoir of EBV virus. In multiple sclerosis, several findings have highlighted the role of plasma cells in secreting and realizing demyelinating antibodies. EBV was replicated in these B cells, but the lytic cycle was not completed without a release of new virions.

In vivo, the primary infection is due in oropharyngeal epithelium in a productive phase, the so-called lytic infection [9,10]. The viral spread is so high and the virus infects the circulating B cells, the viral reservoir, persisting in a quiescent state (latent phase) without a release of infectious particles [11,12,13,14,15]. The biological cycle is similar to other herpesviruses. It is known that this virus infects B-lymphocytes and epithelial cells latently. During this state, the genomic DNA is tethered in a circular episome and only a set of viral proteins are expressed [16,17,18]. Many studies have been identified them as proteins necessary to establish persistent infections in target cells by reducing and inhibiting immune responses in infected individuals. Epstein–Barr nuclear antigens 1 and 2 (EBNA-1 and EBNA-2) are required to replicate the viral genome during the cell cycle progression, using the host DNA polymerase, and to gain a ‘persisting’ state, respectively [19,20,21]. In the last 60 years, several findings have showed that marmoset EBV-infected cells can proliferate and grow in vitro, expressing all of the latent proteins: six nuclear antigens (EBNA-1 to EBNA-6), latent membrane proteins (LMPs), and two non-polyadenylated RNAs (EBERs) [22,23,24,25,26,27,28]. One of them, LMP1, is an integral membrane protein that it mimics the CD40 receptor. The receptor is constitutively activated and this state induces a constitutive proliferation in infected cells. The latent proteins regulate cell cycle progression and apoptosis to avoid all of the pathways related to cellular death mechanisms.

However, EBV can establish a productive state, expressing, more or less, 100 proteins to exert a lytic phase, to generate and produce new viral particles, spreading the virus in many compartments of the body. Several studies have pointed out that the lytic state is related to the severity of EBV-associated cancer or autoimmunity diseases, such as multiple sclerosis (MS) and rheumatoid arthritis (AR). Lytic proteins are classified in three different classes, involving their contributions to the production of virions. Immediate early (IE) proteins are necessary to trigger the viral replication expressing the early and late proteins. The former are required to replicate the viral genome in a defined structure, named concatemers, and released in a nucleocapsid, they gain the plasmatic membrane by a well-defined mechanism [16,17,18,19,20,21,22,23,24,25,26,27,28,29,30]. BFRF1 and BFLF2 encoded two proteins; they are essential to migrate and to translocate the incoming new virions in the cytoplasm by acquiring and losing the nuclear membrane shifts [31,32,33,34]. The cytoplasmic virions get hold of the late proteins that are necessary to structure the tegument and the envelope. DNA recombination techniques have allowed generating, in vitro, several systems to construct a bacterial artificial chromosome (BAC), cloning all the EBV genomes extracted by B95.8 cells [35,36]. This system is used to characterize the function of all the EBV-proteins. BFLF2 knockout mutant was generated to exert the function of this early EBV protein [35]. These studies have shown that this gene encoded a protein involved in DNA viral packaging and in the disabling nuclear egress. The lack of this lytic protein reduced the capsid maturation and gained the released of DNA empty virions, so-called virus like-particles (VLPs)—the major objects of vaccine studies [37,38,39,40,41,42].

In vitro, infected cells switch from latent to lytic phases by trigging the viral replication through several drugs, such as phorbol ester (e.g., TPA-phorbol 12-myristate 13-acetate) and histone deacetylase inhibitors (e.g., sodium butyrate-NaB). The immediate early genes trigger the lytic gene activation cascade, expressing all lytic proteins necessary to structure the virions. BZT, a proteasome inhibitor, also activates the viral replication by exploiting autophagy machinery [43,44,45] (Figure 1).

### 1.2. Kaposi’s Sarcoma-Associated Herpesvirus (KSHV)

Kaposi’s sarcoma-associated herpesvirus (KSHV) is the eighth herpesvirus; it is formally known as human herpesvirus 8 (HHV-8). KSHV was identified by Chang et al. in a biopsy of an HIV-infected and Kaposi’s sarcoma-affected patient. KS was described by a Hungarian clinician named Moritz Kaposi. It was observed in the oldest man from the Mediterranean area. However, KS was detected in the early 1980s in HIV-infected patients. It is also related to primary effusion lymphoma (PEL) and to multicentric Castleman disease (MCD). It infects endothelial cells, monocytes, dendritic cells (DCs), and T lymphocytes. It exerts two phases of infection of EBV: latent and lytic.

The pathogenetic mechanisms underlying the development of this disease are not well understood. Some models have described that the virus expresses during the latent phase; several proteins mimic the cellular protein that usually regulate proliferation and apoptosis by cell cycle checkpoint, or transduction signaling that is activated by several kinases. KSHV latent nuclear antigen (LANA) protein is encoded by the ORF73 gene; it gains viral persistence in the host cells [46,47]. It is involved in dysregulation of cell growth and survival by inhibiting p53 function. LANA can also interact with GSK3β regulating cellular localization of β-catenin, which in turn activates proliferation genes (e.g., cyclin D and c-Myc).

The KSHV lytic state is induced by RTA and K-bZIP proteins; they are the viral homologues of the EBV BRLF1 and BZLF1 immediate and early genes, respectively. As described in EBV, the early and late proteins are necessary to the viral replication and virion maturations [48,49,50,51].

KSHV enhances and promotes cytokine release of IL-6 and IL-10, creating a microenvironment in which the STAT3 transcription factor as well as the NF-kB are constitutively activated [52,53,54,55]. In the model proposed in recent years, KSHV investigators point out the mechanism that involve KSHV in the development of Kaposi’s sarcoma, by monitoring the progression of the disease via several diagnostic molecules. Briefly, it was highlighted that the latent and lytic states work together, leading to an increase of the severity of disease.

PEL patients have unfavorable prognoses and are typically refractory to conventional treatments. Therefore, to find new therapeutical strategies, good understanding of the cellular mechanism (e.g., that it sustains the progression of the disease) is necessary.

The aim of these investigators is to improve the outcome of therapy, exploiting some new drugs, pointing out the cellular and viral pathways involved in these kinds of malignancies.

Many pharmacological drugs inhibit certain pathways, such as apoptosis or autophagy, using in vitro models by enhancing (as well as blocking) the lytic state. In PEL (KSHV-infected) and B95.8 (EBV-infected) cells, it was demonstrated that bortezomib (BZT), approved by the FDA as a drug to treat multiple myeloma patients in 2003, could induce viral replication, regulating autophagy and ubiquitin-proteasome systems [56]. Clinicians should investigate the peculiarity of this treatment, considering it as an adverse effect during therapy. The outcome of the therapy would improve by combining it with other drugs [57,58].

One of the pathways explored by the investigators is autophagy system. This cellular mechanism regulates the protein half-lives, the degradation of damaged organelles (e.g., mitochondria), or the area of endoplasmic reticulum, maintaining cellular homeostasis. Moreover, autophagy is dysregulated in several cancers; sometimes it is linked to chemoresistance. The pathogens are usually degraded by this mechanisms and the functioning of the cell immune systems are often sustained by it. However, γ-herpesviruses exploits the autophagy system, to enhance the maturation and production of new viral particles [59,60] (Figure 1). In the last decade, many pharmacological trials have demonstrated that some molecules, which inhibit autophagy at early or late steps (e.g., chloroquine), in combination with platinum based-cancer therapy, reduce the cancer progression in patients. One must look for a method to deliver these pharmacological drugs without dissolving them into toxic solvents. Several new systems to solve the adverse pharmacological effects induced by these treatments are being planned in nanomedicine and nanodevices.

The first nanosystems have been applied in medicine, to increase the dose and bioavailability of the therapeutics. Currently, nanoparticles (NPs) are known to exert several antiviral mechanisms. Some are required/designed to enhance the efficacy of vaccines.

## 2. Therapeutics Treatments in EBV- and -KSHV-Infected Malignancies

In the last twenty years, investigators and clinicians have studied the molecular mechanisms underlining the pathogenesis of γ-herpesviruses. Several attempts have been made to search for molecules, to design targeted therapies specific to patients—from the bench to the bedside.

Nanosystems could improve standard and targeted therapies, reducing therapeutic toxicity and promoting specific targets as receptors overexpressed in tissue tumors.

All of the studies reported on in this review were collected from PubMed, Google Scholar, and Clinical Trial (https://clinicaltrials.gov/ accessed on 19 October 2021).

### 2.1. EBV-Associated Diseases and Targeted Therapy

Epstein–Barr Virus-associated diseases are related to the activation of cellular and viral mechanisms, as they sustain constitutive proliferations in infected cells. Many studies have improved the standard therapy in EBV-related patients. Latent and lytic phases have been investigated to find viral targets or biomarkers in new therapies, designed for each patient.

RNA-cleaving deoxyribozymes (DNAzymes) (DZ1) are synthetic single-stranded DNA-based catalytic molecules engineered to bind and to cleave target mRNA at specific sites. DZ1 was used as a therapeutic agent in a range of preclinical cancer models; it has entered clinical trials in Europe, China, and Australia [61]. This review surveys regulatory insights into mechanisms of diseases and in the use of catalytic DNA in vitro and in vivo, including nanosensors, nanoflowers, and nanosponges, and the emerging role of adaptive immunity underlying DNAzyme inhibition of cancer growth. This enzyme specifically targets latent LMP1, expressed in most EBV-associated lymphoproliferative diseases and malignancies, and it critically contributes to pathogenesis and disease phenotypes. Thirty years of LMP1 research revealed its high potential as a deregulator of cellular signal transduction pathways, leading to target cell proliferation and the simultaneous subversion to apoptosis mechanism. However, LMP1 has multiple roles beyond cell transformation and immortalization, ranging from cytokine and chemokine induction, immune modulation, the global alteration of gene and microRNA expression patterns to the regulation of tumor angiogenesis, cell–cell contact, cell migration, and invasive growth of tumor cells [62].

It was demonstrated that this EBV protein is also expressed at high levels during viral replication and it is involved in pathogenesis. Patients treated with this antiviral agent reduced the cancer progression in NPC cancer [62]. The lytic phase is a crucial step of the EBV cycle. During this phase, new viral particles are produced and released in the patients. The viral spread in PTDL patients is higher than in healthy patients. This step contributes toward infecting de novo cell hosts, increasing the progression of diseases [62].

In Burkitt’s lymphoma (BL), several anti-viral agents target many proteins expressed during the late or lytic phases. Valproic acid (VPA) or valpromide (VMP), an amide derived of acid valproic, has been demonstrated to prevent the expression of immediate early EBV genes, BZLF1 and BRLF1 lytic genes. These viral proteins are necessary and essential to switch the latent to lytic phase, promoting the transcription of all genes expressed during the productive phase.

Spironolactone, a mineralocorticoid approved for clinical use, inhibits infectious EBV production, by inhibiting the EBV SM protein function. SM protein, phosphorylated by casein kinase II (CKII), is a regulatory early protein in lytic replication of EBV in BL lymphoma [63].

Maribavir (MBV), is approved as a therapeutic to treat human cytomegalovirus (HCMV) infection in allogeneic stem cell and bone marrow transplant recipients, is interesting, because it is also a potent inhibitor of EBV replication [63]. MBV mainly inhibits the enzymatic activity of EBV-encoded protein kinase (EBV-PK), blocking the viral DNA replication, and suppressing EBV lytic gene expression.

Recently, studies have showed that the ephrin receptor A2 (EphA2) is critical for EBV infection of epithelial cells. EphA2 interacted with EBV gH/gL and gB complex, promoting and enhancing EBV internalization in the target cells. The interaction is mediated by the EphA2 extracellular domain (ECD). More studies have identified a small-molecular inhibitor of EphA2, ALW (ALW-II-41–27), strongly inhibiting the proliferation of gastric cancer in vitro and in gastric cancer patient-derived xenografts. These results indicate that specifically inhibiting EphA2 might be an effective strategy in this cancer therapy. The discovery of EphA2 as an EBV epithelial cell receptor has implications for EBV pathogenesis and may uncover new potential targets that can be used for the development of novel therapeutically strategies [64].

All antiviral EBV compounds, as mentioned, are dissolved in toxic solvents. Many clinical studies have demonstrated that the reduction of toxicity is enhanced during patient treatments by monoclonal antibodies. In post-transplant lymphoproliferative disorder (PTDL), CD20 monoclonal antibodies, in combination with bortezomib (BZT), reduce the dosing of standard chemotherapeutics administrated. Rituximab, a first generation chimeric monoclonal antibody targeting the B-cell surface protein CD20, improved the outcomes in CD20-positive non-Hodgkin’s lymphomas (NHLs) and follicle lymphomas. This therapeutic is effective in EBV-positive PTDL because it directly targets CD20-positive cells. Many strategies improve the disease control on the lytic productive phase by EBV-CTLs. Rituximab is more toxic than other drugs. The adverse effects include fevers, rigors, and hypotension. To this aim, to decrease the side effect in patients, new antiviral strategies are being studied, which highlight nanosystem approaches [64].

Nasopharyngeal carcinoma (NPC) is a malignant epithelial tumor affecting nasopharyngeal mucosa. It is different from head and neck cancer, and its pathogenesis is related to multiple etiological factors, such as genetic predisposition, diet, and Epstein–Barr virus (EBV) infection. A therapeutic target in NPC cancer is the latent protein EBV nuclear antigen 1 (EBNA1). This viral molecule is necessary for maintenance, replication, and segregation of viral genome in mitosis during cell cycle progression. EBNA1 binds several cellular molecules, promoting oncogenic transformation of infected cells. A binding DNA domain is located close to the C-terminal. Crystallographic studies of the DNA-binding sequence of the EBNA1 protein revealed that this region has two structural domains: i) a core domain for EBNA1 dimerization (and a sequence-specific DNA interaction); and ii) a flanking domain for DNA contact. Many studies have demonstrated that the combination of four compounds (SC7, SC11, SC19, and SC27) targeting this region reduce EBNA1 activity [64].

Synthesized therapeutics to EBNA1 domains could be a new antiviral strategy, as it could improve the standard therapy in EBV-associated malignancies.

### 2.2. KSHV-Associated Diseases and Targeted Therapy

Several studies have been performed to design pharmacological therapies, to treat EBV- and KSHV-associated malignancies. Several findings reveal research about medications and the key molecules involved in cellular (or in viral) pathways related to pathogenesis and the disease progression of γ-herpesviruses malignancies. Certain molecules inhibit maturation and new viral particle production.

Nucleoside inhibitors of the viral DNA polymerase, such as ganciclovir, cidofovir, foscarnet, brivudine, and adefovir, have been described as potent inhibitors of KSHV replication in tissue culture. In patients, several studies reported that these inhibitors reduced the viral shedding detected by oral samples or by peripheral blood methods. However, in many cases, the DNA polymerase inhibitors are ineffective in the treatment of advanced KSHV-associated diseases [65]. This drug effect is related to the phosphorylation status of these molecules activated by cellular or viral kinases. Viral thymidine kinase (TK), encoded by ORF21, is expressed only in γ-herpesviruses. It is able to activate these drugs during KSHV productive phase by phosphorylation nucleoside analogs. Pyrimidine nucleosides, such as brivudine and azidothymidine (zidovudine, the anti-HIV nucleoside reverse transcriptase inhibitor), are efficiently phosphorylated by KSHV TK [66].

Latent nuclear antigen 1 (LANA1) can bind the KSHV terminal repeat (TR) sequences, regulating viral genome replication, packaging, and segregation during cell cycle progression. Kirsch and colleagues demonstrated that some new small molecules inhibited LANA binding to viral genome. One of these therapeutics, mubritinib (TAK165), was used as a LANA-DNA inhibitor in PEL cell treatments in vivo and in vitro [66,67]. The viral FLICE inhibitory protein (vFLIP) binds the IKK/NEMO complex, regulating, in turn, cell cycle progression and apoptosis. Many studies have solved the vFLIP three-dimensional structure, with the aim to design specific inhibitors in its regulating site. In many studies, an increase of apoptosis in cells treated by Bay 11-7082 was observed—this synthetic drug blockades the vFLIP activation in PEL cells in vitro. In many attempts, RNA interference was used to downregulate the biochemical synthesis of this viral protein to shut down the constitutive expression of the cellular proteins involved in the mitosis and proliferative pathways [66,67]. It is known that the vFLIP complex, with NEMO/IKK and heat shock protein 90 (HSP90), regulate mitogen activated protein kinase (MAPK) phosphorylations. These kinases are involved in K1 latent viral protein activation, regulating the proliferative state of PEL cells.

Several findings, at the preclinical stage, have elucidated the role of γ-herpesvirus productive phase proteins in Kaposi’s sarcoma (KS) pathogenesis [68]. Gallic acid (GA) inhibited ORF50 (RTA), the lytic regulator in the switching from the latent phase, avoiding binding with the RTA response sequences in viral lytic gene promoters [68]. In similar studies, the compound NSC373989 allosteric blocked the quaternary structure of the ORF9/ORF59 complex, inhibiting the viral replication in the concatemeric structure cleaved at TRs during packaging, in the pre-formed nucleocapsids.

Many studies have highlighted that the RNase H-like nucleotidyltransferase domain of HIV integrase had a similar structure to two single-stranded DNA (ssDNA) binding proteins, necessary to trigger the lytic state by activating the viral DNA replication. Raltegravir and dolutegravir, two HIV integrase inhibitors approved for clinical use in HIV patient treatments, were successfully tested in vitro for their inhibition of the KSHV large terminase subunit, encoded by pORF29. KSHV pORF29 is a functional homologue of RNase H-like nucleotidyltransferase in its C-terminal domain. Its inhibition impairs KSHV lytic reactivation in tissue culture [68].

Several studies in mice or in patients demonstrated the role of synthetic inhibitors in cellular mechanisms during the KSHV biological cycle. These compounds usually activate pro-apoptotic mechanisms regulating cell proliferation of host cells infected by KSHV. Many cellular receptors and tyrosine kinases inhibitors, such as vascular endothelial growth factor receptor (VEGFR) and cellular Abelson tyrosine kinases (cAbl), were investigated in several clinical studies. Imatinib, approved to treat patients affected by chronic myelogenous leukemia, was administered to KS patients, resulting in a reduction of lesions. Similar clinical results were observed in a case report study in which a patient treated by sorafenib, a VEGFR inhibitor, showed a complete remission of KS lesions. Many tyrosine kinase inhibitors, such as dasatinib and ponatinib, inhibited the KSHV early gene expression and the productive state in epithelial and endothelial cell lines. Dasatinib regulated endothelial proliferation in KS cancer in a mouse xenograft model [66].

Several clinical attempts were established using the mammalian target of rapamycin inhibitor (mTOR), the key regulator of proliferation and autophagy mechanisms. Sirolimus inhibited cell growth and induced apoptosis in PEL cells, in vitro, and in a mouse xenograft model.

Several therapeutics were used to downregulate constitutive proliferation and apoptosis [66]. BZT is one of the therapeutics potential used to treat patients affected by EBV- and KSHV-associated diseases. Clinicians could consider the γ-herpesviruses productive phase as an adverse effect triggered in these diseases. In vivo, new medications are often related to inhibit autophagy, which is known to be necessary during γ-herpesviruses lytic infection (Figure 2A,B). The reduction of the adverse effects caused by the solvents in which these drugs are diluted would be a good pre-clinical study, in order to propose new treatments.

Monoclonal antibodies are also used in the treatment of KSHV-associated diseases. In the last twenty years, many clinical studies have demonstrated that monoclonal antibodies could be used to perform targeted therapies to improve the outcomes of standard therapies administrated in KSHV-associated malignancies. It was demonstrated that bevacizumab, and humanized monoclonal antibodies, recognize VEGF-A, reduced lesions in 31% of HIV-associated KS patients [66]. Rituximab, an anti-CD20 antibody, administrated in combination with liposomal doxorubicin, was used to treat multicentric Castleman disease (MCD). To exert good therapy response, several therapeutics were mixed with monoclonal antibodies [67,68].

### 2.3. γ-Herpesviruses Associated Disease Treatments the Defeat of Therapeutics: Chemoresistance

Therapeutic regimens for virus-associated lymphomas include radiotherapy, chemotherapy antiviral agents, and targeted therapy. Standard and cytotoxic chemotherapy involves conventional care from most virus-associated lymphomas. Most display a higher rate of chemoresistance compared to solid tumors. The intrinsic and viral components play pivotal roles in the chemoresistant phenotype of cancer cells. Lymphomas associated with EBV infection are usually treated with the same strategy: anthracycline-containing chemotherapy (e.g., cyclophosphamide, hydroxydaunorubicin, vincristine, and prednisone (CHOP)). BL is a mature B-cell NHL described as endemic, sporadic, and immunodeficiency-associated variants. CHOP treatments alone or in combination with methotrexate, cytarabine, ifosfamide, and/or etoposide, are used for patients affected by BL at different stages. In PTDL, CHOP-like therapy is the only approach to treat this disorder [68].

Currently, radiotherapy (RT) is one the therapeutic strategies designed in patients affected by NPC at early stages. During cancer progression, RT is administrated in combination with chemotherapy. In metastatic NPC, the clinicians designed a therapy of a combination of several therapeutics: standard (e.g., gemcitabine plus cisplatin), target molecular therapies (e.g., VEGFR and EGFR inhibitors), and immunotherapies (e.g., immune checkpoint inhibitors) [68].

In KSHV-associated diseases, similar therapeutics are administered to the patients. In KS, the therapeutic strategy dilates the medications with doxorubicin, administrated in many cases by liposomal. This therapeutic binds DNA and inhibits the macromolecule biosynthesis by avoiding the topoisomerase II activation. In AIDS-related KS, the combination of antiretroviral therapy and highly active antiretroviral therapy (HAART) is the first treatment option.

Primary effusion lymphoma (PEL) has poor prognosis, and sometimes the standard therapy is not effective. Usually, the clinicians administer CHOP chemotherapy, and in late stages of the cancer, a combination of medication cocktails contain etoposide and cyclophosphamide.

However, several cellular and viral mechanisms reduce the efficiency of chemotherapeutics and the sensitivity to these anticancer drugs. Usually, these molecules inhibit cellular mechanisms that regulate cell cycle progression and proliferation. This strategy induces apoptosis or autophagy in the host cell infected by these viruses. However, the viral proteins, such as EBV EBNA1-2 and LMP1 or KSHV LANA1-2, activate several molecules that dysregulate the mitosis checkpoints and hijack all of the protein necessary for DNA replication. It was demonstrated that the autophagy degradation mechanism is often elicit, promoting the recycling of macromolecules during cellular stress, such as metabolic dysfunction. Therefore, the infected cell can stand and show chemoresistance to standard medications.

The combination with other therapeutic is crucial to overcome the chemoresistance and sometimes natural compounds or other inhibitors may increase the outcome of the conventional therapy.

The nanosystem may improve the outcome, reducing the toxicity and targeting specific drugs to tissue tumors, avoiding healthy tissue damage [68].

### 2.4. γ-Herpesviruses Associated Disease Treatments and Non-Standard Compounds

Several compounds have been investigated as potential drugs used in combination with standard therapeutics. Some inhibit the viral protein activity and downregulate the proliferative and metabolic pathways, inducing apoptosis in infected cells. In this sub-section, I described the use of many natural compounds in γ-herpesviruses infection.

#### 2.4.1. Quercetin

Quercetin is extracted from flavonol plants and it belongs to the flavonoid group of polyphenols. In an ordinary diet, this natural molecule is contained in many fruits, vegetables, seeds, and leaves. As with other compounds, the bioavailability in humans is very low, because it rapidly clears, with a rate elimination of 1–2 h post-food ingestion. However, these drugs have several anti-inflammatory, antioxidant, and anti-viral properties. Many studies have shown that quercetin treatments reduce ROS levels, stressing p62/SQSTM1, the autophagy adapter protein, and downregulating the expression of heat shock transcription factor (HSF1), and nuclear factor erythroid 2-related factor 2 (NRF2) molecules; they usually sustain this cellular pathway in the PEL cell line [69,70,71].

#### 2.4.2. Curcumin

Similar effects were observed by treatments with curcumin. This natural compound has a bright yellow color; it is extracted from *Curcuma Long L*. (*Zingiberaceae*). It is often contained in food flavoring, food coloring, and cosmetics. It has antiviral effects on herpesvirus infections. As with other drugs, curcumin shuts down viral maturation steps, decreasing the release of new viral particles. In herpes simplex type 2 (HSV2), it regulates the activity of the NF-kB transcription factor, enhancing inhibition of viral DNA replication. In KSHV, ORF50/RTA expression was reduced by curcumin treatments in the BCBL1 cell line [72]. RTA is the lytic master regulator of the lytic state; it triggers the switch from the latent to the productive phase. Its role is also observed in viral genome replication and, thus, it regulates the transcription of lytic genes necessary to capsid formation and maturation in the host cells. Curcumin shuts down the BZLF1 gene in EBV as demonstrated by the CAT assay study on the ZTA promoter [72]. The lack of expression of the ZEBRA encoded protein impaired all of the lytic replications, blocking the productive state of the virus. It is know that curcumin, as a weak and low bioavailability, undermined the antiviral effects of EBV- and KSHV-associated diseases.

#### 2.4.3. Berberine (BBR)

Berberine (molecular formula C_20_H_19_NO_5_ and molecular weight of 353.36) is the main active component of an ancient Chinese herb, *Coptis chinensis Franch*, which has been used to treat diabetes for thousands of years. In the last decade, several findings have demonstrated that Berberine has good antiviral effects on EBV- and -KSHV infections. As described, these viruses sustain a tumor microenvironment by several cytosine releases, such as IL-6 and IL-10. BBR may mitigate this effect, promoting a decrease of latent and lytic protein expression. One target of berberine is the signal transducer and activator of transcription 3 (STAT3), usually constitutively activated in PEL cells lines. It is also known that this natural compound inhibits STAT3 activation and reduces, in turn, the IL-6 release in EBV-positive NPC in vitro [73]. EBNA1 expression is also shut down in NPC, both in vivo and in vitro. During the lytic state, ZEBRA expression is also negatively modulated by the treatments with this natural compound.

In the PEL cell line, as observed by other molecules, BBR targets the NF-kB activation at 30 and 100 μM. These data are confirmed by the xenograft model. NF-kB is a molecule that it is usually activated to sustain the pro-proliferative effect in KSHV PEL cell lines [73].

#### 2.4.4. Melatonin

Melatonin is a neurohormone produced in, and secreted by the pineal gland during the night hours. It is able to move to cerebrospinal fluid (CSF), and is released in the blood, reaching the brain tissue, preventing neuronal damage [74]. Melatonin is also produced by neurons and glia cells. This molecule has been used in treatments related toward inhibiting neurodegenerative disease progression [74]. Melatonin displays anti-inflammatory, antioxidant, and anti-apoptotic properties associated with mitochondria deregulating the cellular bioenergetic system. Many neurodegenerative diseases are due to viral infections (Figure 3A,B). Viruses are able to move toward the blood–brain barrier (BBB), leading to neuronal degeneration [75]. In 1919, the link between the psychosis and influenza disease was discovered during the Spanish flu epidemic [76]. Several findings pointed out the symptoms of PD and Japanese encephalitis virus (JEV). CNS disorder is related to herpesvirus infection, compromising the oligodendrocytes physiology and immune system response. Multiple sclerosis is a chronic autoimmune disease, due to demyelination of neurons resulting in panencephalitis and neuromyelitis optics. The *APOE-e4* allele mutation is linked to the onset of inflammatory responses, as a consequence of herpes simplex type-1 (HSV-1) and human cytomegalovirus (hCMV) infections [77,78,79].

Increased levels of oxygen reactive species (ROS) and antioxidants have been detected in several cancer developments and progressions. Melatonin levels are altered in cancer patients due to a dysfunction in its release. This therapeutic displays anti-tumoral properties, regulating cell cycle progression, apoptosis, induced-oxidative stress, immune stimulation, and growth signaling, exerting anti-proliferative effects [80,81]. It was shown that melatonin represses the Warburg effect, ameliorates disturbed mitochondrial bioenergetics, and is a pro-oxidant in cancer cells, even in cancer stem cells (CSCs) [82,83].

Beclin1 is the key regulator molecule inducing autophagy vesicle systems used to recycle or degrade some cellular organelles, and many portions of endoplasmic reticulum (ER). Melatonin actives or inhibits autophagy in several cancers (Figure 3). HCT116 colorectal cancer cells treated in vitro with melatonin showed cellular senescence and apoptosis, upregulating cleaved effector caspase 3 and LC3-II expression, modulating autophagic-lysosomal vesicle formation [84,85] (Figure 2B). In several cancers, melatonin exerts anti-tumoral effects, shutting down the constitutive proliferative effects by promoting/inhibiting autophagy mechanisms (Figure 3). Cancer cell growth is often regulated by fine mechanisms of balance of autophagy activation or inhibition. It is known that this lysosomal system degrades phosphorylated protein or a part of the cytoplasmic compartment to promote all of the cellular mechanisms that enhance apoptosis.

However, in liver cancer, it was demonstrated that melatonin treatments did not activate the mammalian target of rapamycin complex 1/2 (mTORC1/TORC2) and phosphoinositide 3-kinase class I (PI3K)/protein kinase B (Akt) stimulating autophagy (Figure 2B). The cellular mechanism, enhancing chemotherapeutic resistance and autophagy, sustain the cell cancer proliferation by blocking apoptosis. The stimulated mitophagy reduces in metastatic cancer the damaged mitochondria and restores the redox-metabolic balance in these cancer cells. Usually, mitophagy displays the degradation of damaged mitochondria to impair antioxidant response (Figure 2B) [86,87].

## 3. Nanosystems: From Liposomes to Nanoparticles (NPs)

Infectious disease agents, such as bacteria, viruses, fungi, and parasites, account for approximately 15 million deaths worldwide, with acute respiratory infections and human immunodeficiency virus (HIV) being the leading causes. Good treatments are crucial in the onset and progression of malignancies related to herpesvirus associated-diseases. The phenomena are related to public health, considering the cost of new drugs and new systems to administer the therapeutics, from pre-clinical to clinical purposes. In the last twenty years, nanosystems have been applied to medicines, to find new therapeutic approaches in patient treatments. Second-generation nanosystems were engineered to modulate dose limiting, and to enhance the bioavailability of drugs, such as curcumin and quercetin. This system is also performed to promote delivery to specific tissue sites, to enhance the efficacy of toxic drugs, such as doxorubicin.

In γ-herpesvirus infections, many attempts were made to synthesize vaccines against specific molecules, key regulators of latent or lytic infections.

Organic nanoparticles are the most extensively researched types of nanoparticle for drug delivery and the most widely approved systems for therapeutic use in humans [88].

### 3.1. Polymeric Nanoparticles

Polymeric nanoparticles are colloidal solids with sizes in range from 10 to 1000 nm. The small size help nanoparticles reach tissue cancer by discontinuous vascular endothelial cells and increase the dose of drug delivery to the cells. Polymers approved by the World Health Organization (WHO) and the Food and Drug Administration (FDA) for use in medicine and pharmaceuticals include polylactide (PLA), polyglycolide (PGA), and poly(lactide-co-glycolide) (PLGA). Poly(D,L-lactide-co-glycolide) (PLG) and PLGA-based nanoparticles are most widely used due to their superior biocompatibility and biodegradability profiles. PEG molecules have the capacity to avoid serum protein interaction and to elicit immune system surveillance [89].

#### 3.1.1. Liposomes

Liposomes are spherical carriers, ranging from 20 to 30 nm in size. They are composed of a phospholipid bilayer (which can mimic cell membranes and directly fuse with microbial membranes), containing an aqueous core. Hydrophilic and lipophilic drugs can be incorporated into the inner aqueous cavity or the phospholipid bilayer, respectively. The lipid bilayers display the same properties of the plasma membranes, enhancing the absorption to cell targets. Liposomes have been studied to synthesize vaccines [90].

#### 3.1.2. Inorganic Nanoparticles

Metallic nanoparticles can be smaller than organic nanoparticles, between 1 and 100 nm in size, while their loading efficacies are much higher. There are two main approaches for the synthesis of metallic nanoparticles: the ‘bottom up’ (or self-assembly) approach refers to the construction of the nanoparticle, level-by-level (e.g., atom-by-atom or cluster-by-cluster), and the ‘top–down’ approach uses chemical or physical methods to reduce the inorganic material to its nanosized form. The reaction conditions (pH, temperature, time, or concentration) can be used to modify the nanoparticle characteristics (size and shape), while the reducing agent can influence properties, such as loading capacity, release, and aggregation profiles [91].

#### 3.1.3. Gold Nanoparticles (GNPs)

Gold nanoparticles (GNPs) are widely researched as nanocarriers due to their excellent conductivity, flexibility of surface modification, and biocompatibility methods. Other advantages afforded by their unique physical and chemical properties include gold core (inert and non-toxic) photophysical properties [92].

#### 3.1.4. Silver Nanoparticles (nAg)

Silver nanoparticles are the most effective of the metallic nanoparticles against bacteria, viruses, and other eukaryotic microorganisms, due to the inherent inhibitory and bactericidal potential of silver, and their good conductivity, catalytic properties, and chemical stability. The key mechanisms of action of silver nanoparticles involve the release of silver ions (antimicrobial activity), cell membrane disruption, and DNA damage.

These nanoparticles are an emerging material displaying a large area-to-volume ratio and unique physicochemical properties. The antiviral properties are due to the allosteric interactions between glycoproteins expressed on virus surfaces and the nanoparticles. The positive competition allows manipulating the particle entry and ‘soak up’ on the target cells. They exert the capacity to block DNA viral replication and induce apoptosis or autophagy in the host cells. Silver nanoparticles selectively induce human oncogenic γ-herpesvirus-related cancer cell death through reactivating viral lytic replication [93].

### 3.2. Nanoparticles (NPs) and EBV and KSHV Vaccines

EBV infection is related to the onset of lymphoproliferative malignancy. The increase in viral load is often associated with the severity of the disease compromising the outcome of therapy and the progression of it. Clinicians and investigators have depleted B-lymphocytes, with the aim of reducing viral replication. In observational studies, the use of monoclonal antibodies, such as anti-CD20, expressed (rituximab drug) on mature B cells, reduced the rate of disease progression, related to EBV-infection, by 49% in a historical cohort—18% in the treated group [91]. The mortality ratio was up to 6 months. Autologous EBV-specific T cells have been used to prevent EBV-related lymphoma in PTLDs with high viral loads up to 6 month post-transplantation. Vaccinations, as preventions against developing γ-herpesvirus-associated malignancies, could be good clinical implementations of canonical therapies in Burkitt’s lymphoma or nasopharyngeal carcinoma patients.

In the *Herpesviridae* scientific community, the development of an EBV-vaccine has been debated for several years. The first answer involves the best experimental method to design a vaccine to stimulate and ‘gain’ the activation of an immune response, reducing viral titers in EBV-associated diseases.

VLPs are part of a new strategy developed in the last ten years. Some of them express the gp350 antigen domain, with the aim of recognizing the glycoprotein, by neutralizing antibodies, blocking, in turn, binding with the CD21 B cell receptor [93,94].

gp350 has also been fused with the *Helicobacter pylori* bullfrog hybrid ferritin to generate highly self-assembling nanoparticles. The incorporation of gp350 into ferritin NPs has demonstrated that they enhanced the presentation of the CD21-binding site on the glycoprotein. In vivo, mice vaccinated with these NPs, are protected by EBV-recombinant virus expressing gp350 [95].

Scientist engineered VLPs similar to EBV virions, with the aim of inhibiting the transformation capacities of this virus (Figure 2). They have generated some VLPs, EBV mutant deleted in terminal repeat (TR) sequence required for the DNA viral genome packaging. These VLPs are able to elicit the EBV-humoral and cellular immune response, highlighting its capacity to stimulate the host immune system [96]. However, several pre-clinical studies in animal models have shown that this strategy did not enhance the T CD4+ cells, failing to improve the immune response. Similar findings were showed in the mutant deleted BFLF1–BFRF1A packaging complex—that it led a release of empty capsids without viral genomes [97]. These attempts have indicated to researchers that the best way is to generate/design viral-like particles deleted of latent proteins, such as EBNA1 or EBNA3 [92].

Similar data were obtained in a rabbit model, generating a vaccine to recognize viral proteins, gpK8.1, gB, and gH/gL, into single multivalent KSHV-like particles (KSHV-LPs). Purified KSHV-LPs were similar in size, shape, and morphology to KSHV virions. Vaccination of rabbits with adjuvanted KSHV-LPs generated strong glycoprotein-specific antibody responses; purified immunoglobulins from KSHV-LP-immunized rabbits neutralized KSHV infection in epithelial, endothelial, fibroblast, and B cell lines (60–90% at the highest concentration tested). These findings suggest that KSHV-LPs may be used an ideal platform for developing a safe and effective prophylactic KSHV vaccine [93].

### 3.3. Nanoparticles and Gamma-Herpesviruses Therapeutics

Many natural biomolecules acquire self-assembled lipids, proteins, and polynucleotides. Their discoveries are a starting point to develop and synthetize new material to design nanoparticles (NPs). Nanomedicine is a new branch of medicine that studies nanodevices or nanoparticles, to improve the imaging and acquisition system in diagnosis, as well as in drug delivery in several diseases. The toxicity of these materials has led to ethical debates. According to researchers, inhaling these nanoparticles is considered very dangerous. Thus, safety standard procedures have been approved by ethics and technical committees.

NPs come in different sizes, shapes, and surface molecules (e.g., peptides are acquired in a tridimensional structure, working as good receptors for ligands expressed by specific cells), with peculiar properties used in cancer treatments (Figure 4A). The common nanoparticles are liposomes. In the 1960s, researchers used them as carriers for some drugs based on knowledge in these fields. However, they were not suitable for use for clinical aims. Currently, NPs are known to exert their antiviral activities by several mechanisms. They are engineered to have small particle sizes, used towards specific tissue sites. The large surface area to volume ratio ensures that the binding site has the right structure to accommodate large drugs or therapeutics [94]. Finally, they have tunable surface charges that enhance the negatively charged cellular membrane [94] (Figure 4A). Silver nanoparticles and dendrimers have intrinsic biomimetic properties, acquiring anti-viral effects in host cells. Sometimes, NPs are covered by stable structures (polietilenglicol) (PEG), increasing optimized drug dosing, improving delivery and therapeutic retention times (Figure 4A) [95,98,99]. Some of these NPs are designed to move to the blood–brain barrier (BBB). This site is not reach by conventional therapeutics and NPs designed to cross the BBB, leading to a shutdown of viral replication and viral load [100].

NPs are designed to drug and gene delivery, using fluorescent biological labels, detection of proteins, pathogens, and tumors, separation and purification of biological molecules and cells, tissue engineering, and MRI contrast supported by pharmacokinetic studies. However, some nanoformulations have helped with overcoming the problems related to drug solubility and bioavailability, acting as antiviral agent deliveries in several systems.

To increase adsorption of natural compounds, such as curcumin, researchers have designed nanoparticles that are able to increase bioavailability to the target site. As described in many studies, curcumin is carried by nanoparticles and solid-lipid nanoparticles.

The goal is to design a nanostructure that enables the delivery, the recognition from the immune system (immunogenicity), the safety, the bioavailability, and the stability of NPs in the blood stream.

Lipid-based nanoformulation is designed as a carrier for antiviral compounds. In contrast to polymers, lipids have the peculiarity of being inert, have low toxicity, are immunogenic, and are smaller. Some liposomes employ spherical structures, and are generated by the use of phospholipids, which entrap hydrophilic (as well as hydrophobic) drugs [96,97,101,102,103]. Their molecular formulations are similar to double lipid bilayer biological membranes, i.e., this mimic their structures. The lipid layers of liposomes protect the drug from gastrointestinal degradation and help sustain drug release (Figure 4B). This method has improved the oral rather than the parental administration. However, in vitro studies have demonstrated that their use is restricted to low-drug loading and physical instability.

## 4. Conclusions

In the proposed review, novel therapeutic strategies to improve the outcomes of several malignancies in EBV- and KSKV-infected patients were summarized.

Several studies have shown their roles in many diseases, such as cancer and immunologic disorders. Proliferative pathways, regulated by the progression of cell cycle, apoptosis, and autophagy mechanisms, are often malfunctioned by overexpression of key master kinases. Autophagy recycling and degradation systems maintain cellular homeostasis and protein turnover. Aminoacidic deprivation and serum starvation induced a cellular stress, counteracted by the expression of proliferative molecules in balance mechanisms active by bioenergetic status. Xerophagy is exploited by the cell host to activate the immune system, to avoid and destroy microorganism infections. However, γ-herpesviruses hijack autophagic vesicles, enhancing EBV and KSHV virion maturation, and in turn, inhibit the last step of this lysosomal system. Many attempts by scientists and clinicians will improve the outcomes of standard and targeted therapies. In the last decade, many therapeutics have been designed to target specific cellular and viral proteins with the aim of changing the microenvironment, sustaining the proliferation and metabolic state of infected cells. Some of them are natural compounds. Researchers have conducted many studies, performing new therapeutic strategies, using natural drugs to enhance the patient responses to standard therapies. CRISP Cas9 and RNA interference were performed with the aim to shut down the expression of several proteins, such as receptors or kinases, which in turn regulate the cell cycle progression. However, these methods show some limitations. Natural compounds have very low bioavailability. This effect reduces its antiviral properties. Several standard and targeted therapeutics induce chemoresistance.

Nanotechnologies are applied to medicine, ameliorating the imaging diagnosis and the drug delivery of many therapeutics. NPs are engineered with the aim to limit doses of toxicity and to improve drug delivery to specific tissue areas. It is known that the endothelial cells of blood vessels are usually discontinuous in areas close to cancer tissue. Therapeutics encapsulated in nanoparticles, covered by specific ligands, reach the tumor areas, and bind the receptors expressed by cancer cells. In many studies, scientists pointed out the overexpression of receptors, such as Her2 in triple negative breast cancer. This biological effect is exploited by clinicians, to treat the patients using targeted therapies. The drugs are delivered on the tumor area and the adverse effects are reduced in the healthy tissue.

VLPs are also synthesized to expose several viral epitopes on their surfaces, eliciting ‘immune system promoting’ by recognizing viral glycoproteins in EBV- and KSHV-associated diseases.

NPs have scientific relevance in virologic fields; they have also the capacity to improve vaccine production or drug delivery.

## Figures and Tables

**Figure 1 ijms-22-11407-f001:**
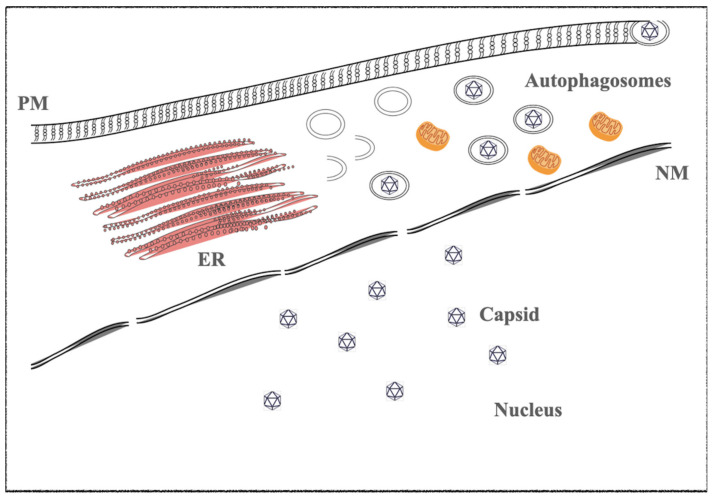
Proposed model of a γ-herpesviruses maturation in an EBV- and KSHV-infected cell line on several stimuli. The figure highlights the viral steps during the productive state. The capsid and viral DNA is packaged in the nucleus of the infected cell host. The capsid acquires the primary envelope at the nuclear membrane (NM) and gains the cytoplasmic compartment. During maturation, they are moved to the plasmatic membrane by autophagic double membrane vesicles (autophagosomes). Autophagy machinery was hijacked by herpesviruses to acquire the last viral envelope surrounded by several glycoproteins. NM—nuclear membrane; PM—plasma membrane; and ER—endoplasmic reticulum, as indicated in the figure.

**Figure 2 ijms-22-11407-f002:**
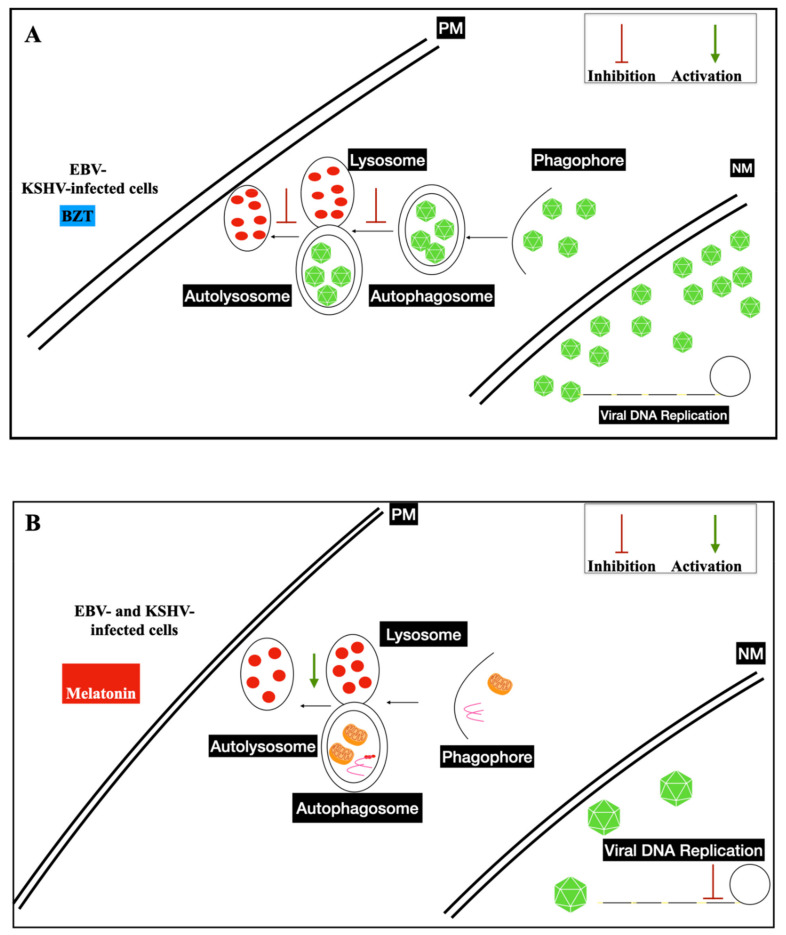
Proposed model of γ-herpesvirus maturation stimulated by several therapeutics. (**A**) Figure of EBV- and KSHV-infected cells and productive lytic cycles hijacking autophagosome double membrane vesicles during virion intracellular maturation triggered by BZT (bortezomib). (**B**) As shown in Figure 2A, EBV and the KSHV lytic cycle are inhibited by melatonin treatments, displaying their anti-viral properties. Autophagy machinery is activated by therapeutics, as shown by autolysosome (autolysosomes) vesicle formation. They enter function recycling or degrade the cellular substrates. During DNA viral replication, the concatemers show that the terminal repeats (TRs) (yellow lines) are essential and necessary for the packaging steps. The blunted end line (red) and arrow (green) indicate the inhibition and the activation of viral or cellular mechanisms, respectively. NM—nuclear membrane and PM—plasmatic membrane are indicated in the figure.

**Figure 3 ijms-22-11407-f003:**
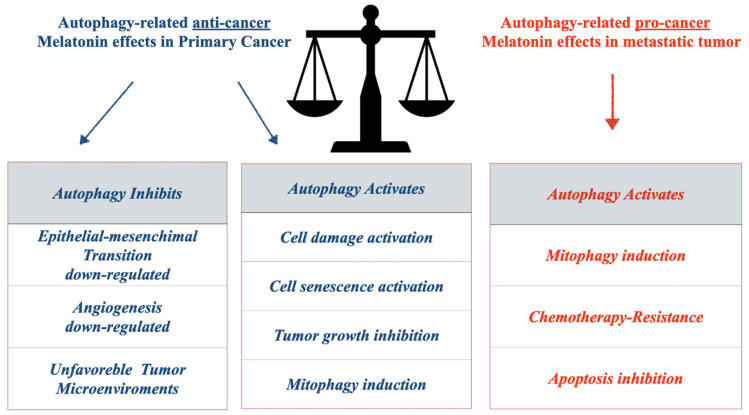
Summary of pro- or anti-cancer melatonin activities related to autophagy. In the schematic tables, it is summarized the effect of melanin during cancer progression. The anti-cancer (blue) and pro-cancer properties (red) describe the melatonin effects, in the tables reported in the picture.

**Figure 4 ijms-22-11407-f004:**
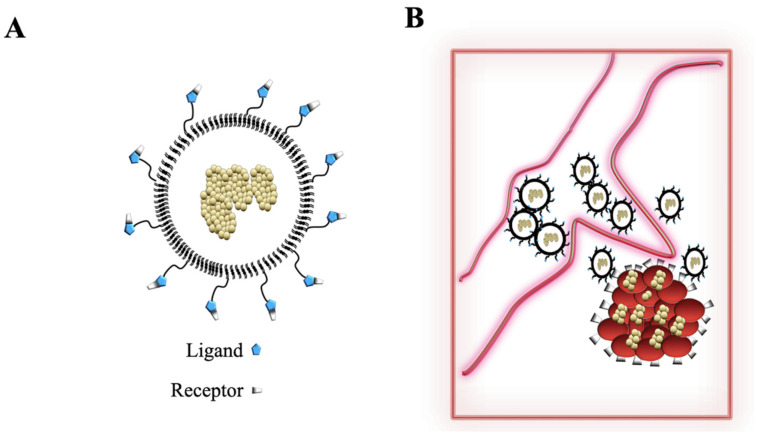
Nanoparticles (NPs) engineered as liposomes covered by polietilenglicol. (**A**) Nanoparticle figures. They are synthesized, to be covered by ligands (blue) interacting with specific receptors (white and black) expressed on tumoral cells. The therapeutics are encapsulated inside the liposomes. (**B**) NPs deliver the drug to the cancer tissue and they bind specific receptors. Therapeutic molecules are realized and they promote apoptosis in cancer cells.

## Data Availability

All of the studies reported on in this review were collected from PubMed, Google Scholar, and Clinical Trial (https://clinicaltrials.gov/ accessed on 19 October 2021).

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
