# Peer review of "Nanotechnology Frontiers in γ-Herpesviruses Treatments"

_ijms, 2021, doi:10.3390/ijms222111407_

Round 1
Reviewer 1 Report
Although the manuscript presents a good topic, there is a need for substantial improvement, as well as some critical concerns and comments, are required to be clarified and addressed.
- Unfortunately, the manuscript is not well-organized, where the presented sections are not correlated, leading to obvious poor quality. Only section 1 and its subsections are the good parts of the manuscript, where the author pointed out the general background on gamma-herpesviruses. The other parts should be reorganized.
- Section 2 (2. Therapeutics treatments in EBV- and KSHV-infected patients) and its subsections should be removed and replaced with a section that should cover the following information:
- A) The currently used and approved pharmacological medications for clinical use (such as acyclovir and related nucleoside analogs) and their limitations (for example, the problem of drug resistance, and others).
- B) Potential drugs that have been tested preclinically (in vitro and in vivo), such as natural and synthetic drugs, and their limitations.
- C) The reason behind the need for new treatment options such as nanotechnology approaches. All these points should be written concisely without broader details.
I recommend the author some references that could be used regarding the natural drugs (DOI: 10.3390/microorganisms9020292) and (DOI: 10.3390/v13061014).
3. It would be better to add information about the period covered by this paper. This is to ensure that the paper is up to date.
4. Information about the used databases for collecting and or extracting the data. should be highlighted.
5. Section (3. Nanosystem in γ-Herpesviruses infection). This section is a very important part of the manuscript since it reflects the main focus of the manuscript. This section should be re-written with deeper details that cover recent advances in nanotechnology techniques/strategies that might be used in the prevention of treatment of gamma-herpesvirus infections. Additionally, all available mechanisms, including molecular or cellular ones should be covered and highlighted.
6. The abstract and conclusion sections should be revised according to the newly added information.
7. Also, I recommend the author add another section that provides new insights into the future of nanotechnology approaches for the prevention and therapy of gamma-herpesvirus infections.
8. Finally, I recommend the author double-check the whole manuscript for grammatical and typing errors.
Author Response
Reviewer 1.
Q1. Section 2 (2. Therapeutics treatments in EBV- and KSHV-infected patients) and its subsections should be removed and replaced with a section that should cover the following information:
A)The currently used and approved pharmacological medications for clinical use (such as acyclovir and related nucleoside analogs) and their limitations (for example, the problem of drug resistance, and others).
B) Potential drugs that have been tested pre-clinically (in vitro and in vivo), such as natural and synthetic drugs, and their limitations.
C) The reason behind the need for new treatment options such as nanotechnology approaches. All these points should be written concisely without broader details.
A1. Thanks for a good suggestions. The section 2 and 3 are re-written and all some detail are added.
Q2.I recommend the author some references that could be used regarding the natural drugs (DOI: 10.3390/microorganisms9020292) and (DOI: 10.3390/v13061014).
A2. Two references is added in the manuscript.
Q3. Section (3 Nanosystem in γ-Herpesviruses infection). This section is a very important part of the manuscript since it reflects the main focus of the manuscript. This section should be re-written with deeper details that cover recent advances in nanotechnology techniques/strategies that might be used in the prevention of treatment of gamma-herpesvirus infections. Additionally, all available mechanisms, including molecular or cellular ones should be covered and highlighted.
A3. New data are added to the section 3.
Q4. The abstract and conclusion sections should be revised according to the newly added information.
A4. The abstract and the conclusion have been revised
Q6. Also, I recommend the author add another section that provides new insights into the future of nanotechnology approaches for the prevention and therapy of gamma-herpesvirus infections.
A6. I explained these perspectives in the conclusion
Q7.Finally, I recommend the author double-check the whole manuscript for grammatical and typing errors.
A7. The typing mistakes are checked in the manuscript.

Reviewer 2 Report
The review entitled “Nanotechnology frontiers in gamma-Herpesvirus treatments” by G. Marisa focuses on recent advances in nanomedicine in treatments of diseases caused by infection with Epstein-Barr virus (EBV) and Kaposi’s Sarcoma-associated Herpesvirus (KSHV). The review emphasizes recent advances in therapies, particularly pharmacological inhibitors, and which signaling pathways and networks are targeted by them. Additionally, the author further details how nanomedicine, namely nanoparticles (NPs), are used to limit toxicity and off-target effects of pharmacological inhibitors.
Overall this is an informative and timely review, and my recommendations for improvement are primarily centered on the grammatical inconsistencies. The author of this review has an excellent grasp of the literature. Still, more careful attention to sentence structure, consistent naming and use of abbreviations, and pluralization of terms is needed to make the review more readable. The scientific content of the study is relatively high; it just needs more polishing.
Major comments:
- Figure 1 is offset in the pdf and needs to be centered on seeing the whole image.
- The use of gamma-herpesvirus or γ-herpesvirus must be consistent throughout the review. The same comment for nanoparticles is sometimes abbreviated NPs and sometimes spelled out in its entirety, as well as for PEG.
- While I love Figure 3A, I’d request that the coloring scheme be different for the Red/Green colorblind. Please consider using blue instead of green.
Minor comments.
- There are many grammatical inconsistencies scattered throughout the review. This list is not all of them, but some that are notable.
- In paragraph 2, please change “In this review, we will discussed about” to “In this review, we will discuss….”
- The following sentence needs to be rewritten: One of them, LMP1, is an integral membrane protein that mimics the CD40 receptor constitutively activated to induce proliferation in infected cells.
- The “s” in the following sentence needs to be added onto the end of “compartment”: …viral particles spreading the virus in some compartments of the body.
- The word “study” needs to be changed to the plural “studies” in the following sentence as the author is referencing multiple independent manuscripts: “The study have been showed that this gene…”
- The word “is” needs to be changed to “was” in the following sentence. It references previous work: KSHV is identified by Chang et al. by detecting it in biopsy of HIV-infected and Kaposis’s Sarcoma-affected patient.
- The following sentence needs to be reworded: It is observed in the oldest man in the Mediterranean area but in the early 1980s it was associated to HIV-infected patients.
- The word “state” needs to be changed to the plural “states” in the following sentence: It exerts two state of infection as EBV: latent and lytic phase.
- The following sentence needs to be removed: Several pathways are constitutively activated.
- I recommend restructuring the following sentence: “In vitro models, many pharmacological drugs inhibit some pathways such as apoptosis or autophagy…” to “Many pharmacological drugs inhibit some pathways such as apoptosis or autophagy using in vitro models…”
- The word “halftimes” needs to be changed to “half-lives” in the following sentence: This cellular mechanism regulates the proteins halftimes.
- The following sentence needs to be restructured: In the last decade, many pharmacological trials have been demonstrated that some molecules, that inhibited autophagy at early or late steps (e.g., chloroquine) in combination with platinum base-cancer therapy, have been shown to take an advantage in the patients treatments.
- The following sentence needs to be rewritten: EBV replicated in these B cells, but the lytic cycle wasn’t complete without a way not of nee visions in the patients.
- The word they needs to be removed from the following sentence “VLPs is a new strategy that they have been developed in the last ten years.”
- The following sentence needs to be restructured: “The inhalations of nanoparticles that don’t exist in nature is very dangerous for the scientists that they worked out to produce them.”
- There is a missing reference in the sentence: In the last decade, EBV infection is also related to the onset of immunological disorders such as multiple sclerosis and rheumatoid arthritis (REF).
- I think the author meant to use the word “ubiquitinated” instead of “uniquitinated” in the following sentence: BZT inhibits proteasome avoiding the ubiquitinated proteins…
- Some of the references are in boldened brackets, and some are not.
Author Response
Reviewer 2
Q1.Figure 1 is offset in the pdf and needs to be centered on seeing the whole image.
A1. The figure 1 is centered in the page.
Q2.The use of gamma-herpesvirus or γ-herpesvirus must be consistent throughout the review. The same comment for nanoparticles is sometimes abbreviated NPs and sometimes spelled out in its entirety, as well as for PEG.
A2. It is correct.
Q3. While I love Figure 3A, I’d request that the coloring scheme be different for the Red/Green colorblind. Please consider using blue instead of green.
A3. I have changed the color.
Q4. Minor Comments. There are many grammatical inconsistencies scattered throughout the review.
A4. All the typing mistakes and re-structured sentences are now corrected.

Round 2
Reviewer 1 Report
The manuscript has been significantly improved.
Author Response
Answer to Reviewer 1R2.
Q1. The authors replied to the comments
A1. It’s correct.
